

# Open-source software library for real-time inertial measurement unit data-based inverse kinematics using OpenSim

Jere Lavikainen, Paavo Vartiainen, Lauri Stenroth and
Pasi A. Karjalainen

Department of Technical Physics, University of Eastern Finland, Kuopio, Finland

## ABSTRACT

**Background:** Inertial measurements (IMUs) facilitate the measurement of human motion outside the motion laboratory. A commonly used open-source software for musculoskeletal simulation and analysis of human motion, OpenSim, includes a tool to enable kinematics analysis of IMU data. However, it only enables offline analysis, *i. e.*, analysis after the data has been collected. Extending OpenSim's functionality to allow real-time kinematics analysis would allow real-time feedback for the subject during the measurement session and has uses in *e.g.*, rehabilitation, robotics, and ergonomics.

**Methods:** We developed an open-source software library for real-time inverse kinematics (IK) analysis of IMU data using OpenSim. The software library reads data from IMUs and uses multithreading for concurrent calculation of IK. Its operation delays and throughputs were measured with a varying number of IMUs and parallel computing IK threads using two different musculoskeletal models, one a lower-body and torso model and the other a full-body model. We published the code under an open-source license on GitHub.

**Results:** A standard desktop computer calculated full-body inverse kinematics from treadmill walking at 1.5 m/s with data from 12 IMUs in real-time with a mean delay below 55 ms and reached a throughput of more than 90 samples per second. A laptop computer had similar delays and reached a throughput above 60 samples per second with treadmill walking. Minimal walking kinematics, motion of lower extremities and torso, were calculated from treadmill walking data in real-time with a throughput of 130 samples per second on the laptop and 180 samples per second on the desktop computer, with approximately half the delay of full-body kinematics.

**Conclusions:** The software library enabled real-time inverse kinematical analysis with different numbers of IMUs and customizable musculoskeletal models. The performance results show that subject-specific full-body motion analysis is feasible in real-time, while a laptop computer and IMUs allowed the use of the method outside the motion laboratory.

Corresponding author
Jere Lavikainen,
jere.lavikainen@uef.fi

## INTRODUCTION

Inertial measurement units (IMUs) are measurement devices that contain triaxial magnetometers, gyroscopes, and accelerometers. IMUs used in biomechanics are usually packed into cases that fit on a human palm. They utilize sensor fusion algorithms such as Kalman filters to estimate the three-dimensional orientation of the IMUs in space (*Paulich et al., 2018*). This information can be used as an alternative to marker-based optical motion tracking systems to perform analysis of human movement. Compared with optical motion tracking systems, IMUs are cheaper, can be attached to the subject without the palpation of anatomical landmarks, do not suffer from line-of-sight issues, are not limited to a specific target volume and can be used in field conditions. These advantages come at a small cost of accuracy compared with optical motion tracking systems (*e.g.*, joint angle errors in the lower limbs are generally between 0 and 15 degrees but vary strongly per joint and motion type) (*Poitras et al., 2019*), and IMU-specific error sources such as drifting (*Saber-Sheikh et al., 2010*). In addition, IMUs can be coupled with electromyography (EMG) electrodes to further enhances the versatility of these sensors for analyzing human movement in sports and clinical applications (*e.g.*, Cometa Srl, Cometa Systems | Wireless EMG and IMU Solutions, https://www.cometasystems.com/; Delsys Incorporated, Trigno® Avanti Platform—Delsys, https://www.delsys.com/trigno/; Noraxon USA, Ultium EMG | Noraxon USA, https://www.noraxon.com/our-products/ultium-emg/).

Analysis of kinematics of motion is typically done offline after measurement and data collection in a process called inverse kinematics (IK). Recent years have seen progress in some real-time IK (RTIK) analysis solutions and systems, but these studies (*Bonnet et al., 2013*; *Borbély & Szolgay, 2017*; *Falisse et al., 2018*; *Miezal, Taetz & Bleser, 2017*; *van den Bogert et al., 2013*; *Yi et al., 2021*) have mostly focused on specific marker sets and models and their generalization to arbitrary measurement setups is difficult. For example, in real-time IMU-based applications, *Bonnet et al. (2013)* estimated the RTIK of the trunk and lower limbs using a single IMU located at the lower back. In another example, sensors containing IMUs and EMG electrodes, which were placed on surface musculature, were used in a recent study (*Yi et al., 2021*) to calculate the real-time kinematics and kinetics of the lower limb. However, these studies rely on complex computational methods, making it difficult for others to repeat and adapt the experiment without knowledge of sensor fusion or deep learning. The IMU-based RTIK solution by *Miezal, Taetz & Bleser (2017)* does not allow easy switching between musculoskeletal models or quantify how real-time the solution is.

It has been shown that a full biomechanical analysis of joint and muscle function can be obtained in real-time with a C/C++ software library (*van den Bogert et al., 2013*). The software library was capable of reading marker data and performing IK and inverse dynamics on a full-body model at more than 120 samples per second. However, the software is commercial and relies on a single predefined model rather than a subject-specific or user-defined musculoskeletal model, limiting its usefulness in research. Additionally, *Falisse et al. (2018)* showed that comparing its outputs with those of another similar software (OpenSim 3.3) resulted in statistically significant differences in joint

kinematics, kinetics, and muscle forces, highlighting the dependency of the output on the selected model. Hence, it is invaluable that the user can select or generate a model that best fits to the application or research question. The OpenSim software for analyzing the kinematics and dynamics of musculoskeletal systems (*Delp et al., 2007*) offers a solution to the aforementioned issues because it is free and open source, it has a graphical user interface, and it works with customizable musculoskeletal models. While its IK algorithm originally utilized only marker-based motion capture data, since version 4.1 (*Seth et al., 2018*) it has been possible to utilize IMU orientations as input to the IK algorithm of OpenSim to solve skeletal motion (*i.e.*, joint angles). The IMU orientation-based IK algorithm minimizes the sum of squares of the difference between experimental IMU orientations and corresponding segment orientations of the model. Other software with capabilities similar to OpenSim (*Damsgaard et al., 2006*) exist, but the fact that OpenSim is open-source makes it readily available to anyone, enables a variety of community-made modifications and add-ons, and enables the user to view the source code to better understand and troubleshoot the workings of the software. Its customizable models allow the creation of personalized bony geometry, *e.g.*, from imaging data (*Valente et al., 2017*), and the inclusion of muscles in the models allows retrieval of muscle length and similar data that can be used in further analyses. Therefore, although other software may offer kinematics or kinetics based on biomechanical models, enabling real-time analysis with OpenSim has advantages in verifiable and customizable motion analysis and research. While OpenSim has been used for marker-based data to calculate inverse kinematics and inverse dynamics of human motion in real-time (*Pizzolato et al., 2017*), OpenSim-based real-time calculation of IMU-based kinematics with open-source code would enable others to adapt the solution for customizable motion analysis in portable settings.

IMU-based RTIK solutions that utilize the OpenSim API have been recently developed by *Stanev et al. (2021)* and *Slade et al. (2022)*. *Slade et al. (2022)* developed and tested an open-source IMU-based IK system for a microcontroller that can be carried on the subject. Their implementation uses a simplified musculoskeletal model and relaxed IK error tolerance to enable real-time IK with the limited computational power of the microcontroller. With the full computational capacity of the microcontroller (*i.e.*, four threads), they could calculate full-body IK at a throughput of approximately 20 operations per second and at a delay of approximately 200 ms. *Stanev et al. (2021)* have published an open-source software framework that allows kinematical and dynamical analysis of motion. Their software performs the analysis in real-time and supports both IMU- and marker-based data.

The aim of this study was to develop a freely available software library that reads orientation data from IMUs and calculates the IK on a user-given musculoskeletal model in real-time using OpenSim 4.1 API. The development of this work was done prior to the publication of works by *Slade et al. (2022)* and *Stanev et al. (2021)*. Since release of the aforementioned software, our work is not novel but represents an alternative implementation to solve the same problem. The development occurred independently of these software and may therefore provide the community additional value to these previous implementations. To assess the performance of our software we quantified the

software's IK execution time and throughput with different numbers of processor threads calculating the IK and different numbers of IMUs, and determined if lowered input data frequency resulting from live visualization meaningfully affects calculated ranges of motion (ROMs).

## MATERIALS AND METHODS

### Working principles of the software

A software library for reading real-time IMU orientation data as quaternions and processing the data to calculate the IK of a musculoskeletal model was developed using C++ and published on GitHub (https://github.com/jerela/OpenSimLive). The software utilizes OpenSim 4.1 API to invoke methods that calibrate the musculoskeletal model and perform the IK to solve joint angles using quaternion-based orientation data from live measurements.

For each time point, the IK algorithm of OpenSim uses orientation information from all IMUs to find the poses of individual bodies of the musculoskeletal model that, in the least squares sense, minimize the error between experimental IMU orientations and the orientations of the corresponding bodies. For information about the mathematics behind the IK algorithm, see Delp et al. (2007), where it is explained for markers instead of IMUs; for IMUs, IMU orientation error is minimized instead of marker coordinate error. OpenSim's IK algorithm for IMU data is briefly presented in Supplemental Materials, although we did not modify it.

The software library supports Xsens MTw Awinda™ (Xsens Technologies B.V., Enschede, Netherlands) and Delsys Trigno® (Delsys Inc., Natick, Massachusetts, USA) IMUs. The open-source nature of the software library allows others to add support for other devices. IMU orientations are received wirelessly as quaternions using the Xsens Device API or individual quaternion elements are read from a byte stream via socket communication sent by Trigno Control Utility. Information about which IMU corresponds to which body on the musculoskeletal model is read from an XML file. Instead of reading orientation data from the actual IMUs, an option to generate randomized quaternion orientations for testing purposes without IMUs is available.

The orientation information from IMUs is combined in a time series table that contains only one sample, i.e., time point. The time series table is given to OpenSim's IK solver object, which solves the IK for that time point. The process is repeated for each sample. The resulting time series of joint angles can be saved in a text file in .mot format, which allows the output to be viewed using the OpenSim graphical user interface. The read quaternions, and also EMG time series in the case of Trigno® Avanti sensors, can be saved in a text file for later offline analysis.

The working principle of the software library is straightforward. Producer-consumer thread synchronization is used to get orientation data from IMUs. A producer thread and a consumer thread run concurrently. The producer reads orientation data as quaternions from the IMUs and saves it into a buffer that is shared between the threads. The consumer reads and removes data from the buffer and assigns an IK task to a thread in a thread pool. This way there can be several concurrent IK calculations, improving the throughput of the

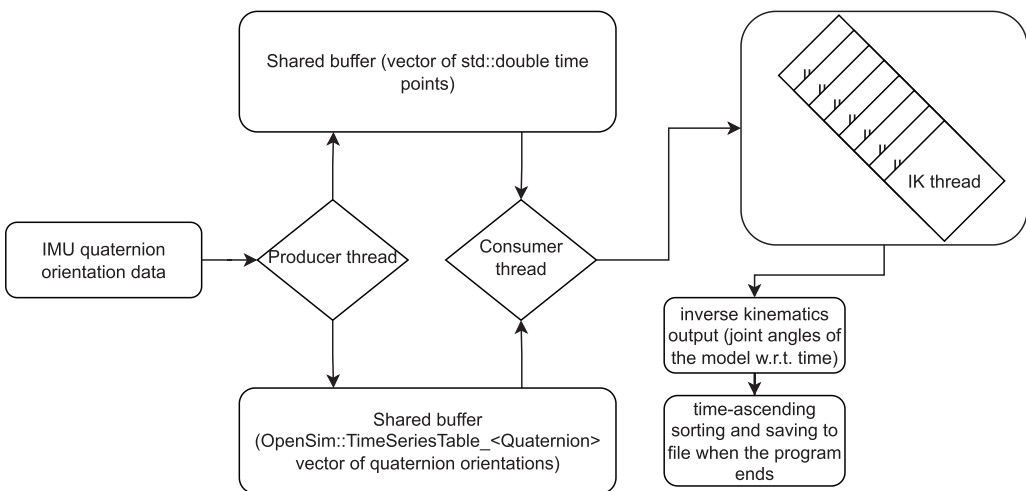

**Figure 1 A diagram illustrating the working principle of the inverse kinematics (IK) workflow.** Orientation data of inertial measurement units (IMUs) is read as quaternions by the producer thread and saved to a buffer. Time values are saved to another buffer. The consumer thread reads data from both buffers and initiates new threads that calculate IK based on the data. IK threads output joint angle values for the model. Within an IK thread, the IK output can be sent to a visualizer window. The visualization is based on the Simbody (*i.e.*, the physics engine used by OpenSim) API and not a part of our software library and hence not described here in more detail. When the program finishes, the IK output frames can be sorted in a time-ascending order and saved to file.

program. The maximum number of concurrent IK threads is defined by the user. If that number is already active when the consumer thread starts another IK thread, the consumer thread will wait for one of the IK threads in the thread pool to finish. A diagram illustrating the workflow is shown in Fig. 1.

The software library has been tested to work on 64-bit Windows 7 and Windows 10 operating systems. The source code of the software library is available on GitHub at https://github.com/jerela/OpenSimLive.

## Experimental data

Performance tests of the software library and error comparison of joint angles between real-time IK and offline IK were done for walking data of a single subject. The subject was of legal age, had no known musculoskeletal disorders or other conditions that affect gait, and gave their written consent to participate in the study. A total of 12 IMUs (Xsens MTw Awinda, Enschede, Netherlands) were strapped on the subject's upper arms, forearms, chest, pelvis, thighs, shanks, and feet, as shown in Supplemental Information. We recorded 10 trials while the subject walked on an instrumented treadmill (Motek Medical B.V., Amsterdam, Netherlands) at a speed of 1.5 m/s (5.4 km/h) and the IMUs transmitted their orientations at a sample rate of 60 Hz. Each trial contained approximately a minute of gait data. The subject was instructed to take the standard anatomical position at the beginning of each trial to calibrate the IMUs on the musculoskeletal model as per the standard IMU calibration procedure of OpenSim (elaborated in Supplemental Materials). When the subject was in the desired calibration pose, the user of our software library pressed a key to

**Table 1 Conducted performance tests, the parameters that were varied in them, and their purposes.**

| Test | Varied parameters | Purpose |
|---|---|---|
| Execution times with computer-generated random unit quaternions | Computer, MS model, number of IMUs | Benchmark upper limit of execution times |
| Execution times with gait data | Computer, MS model, number of IMUs | Quantify execution times in normal use scenario |
| Throughputs with gait data | Computer, MS model, number of IK threads, number of IMUs | Quantify throughput in normal use scenario |

Note:
IMU, inertial measurement unit; MS, musculoskeletal.

calibrate the IMUs. No model scaling was done because we tracked only sensor orientations which are independent of model dimensions.

RTIK was calculated during the measurements. The Simbody visualization of RTIK was enabled, which reduced the throughput of RTIK from the 60 Hz sampling frequency of the IMUs to 45 Hz on average on the desktop computer. This RTIK output was saved to file at the end of each trial. Additionally, the time series of received IMU orientations as quaternions were saved to a file at the full 60 Hz sampling frequency after each trial; the time series were used in calculating execution times and throughputs of the software library.

## Performance tests

To quantify the performance of the software library, we conducted performance tests using computer-generated IMU orientations and IMU orientations from recorded human walking (Table 1). Two performance measures were quantified: throughput and execution times of IK. The throughput describes how many IK operations are calculated per second on average when the communication between the producer and the consumer threads is included in the operation. The execution times describe the delays that the calculation of IK for a single time frame takes, *i.e.*, how long it takes after retrieving IMU orientations from one time point to retrieve the corresponding joint angles of the musculoskeletal model. Therefore, throughputs are increased by using multiple IK threads but execution times are not. To retrieve time points, the std::chrono::high_resolution_clock class was used in the C++ implementation.

The tests were conducted with two musculoskeletal models, the Gait2392 lower extremities and torso model (23 degrees of freedom, DOFs; referenced from here on as the lower body model) (*Anderson & Pandy, 1999*, *2001*; *Delp et al., 1990*; *Yamaguchi & Zajac, 1989*) and the Hamner full-body model (29 DOFs, referenced from here on as the full-body model) (*Hamner, Seth & Delp, 2010*). In tests involving the lower body model, data from one and seven IMUs were used; with the full-body model, data from one, seven, and 12 IMUs were used (Table 2). All joint angles that were unlocked in the model by default were solved by the IK algorithm of OpenSim, but meaningful results were obtained only for joint angles defined by the available IMU data, such as joint angles between two segments that both had an attached IMU sensor. Finally, the tests were conducted with two computers, a laptop (HP EliteBook 8570w: Windows 10 Education 64-bit, Intel Core i7-3740QM 2.70 GHz 8-CPU processor, 8192 MB RAM) and a desktop (Fujitsu Celsius

**Table 2 The MS model segments whose inverse kinematics were calculated during performance tests, presented by the number of IMUs whose data was utilized.**

| Number of IMUs | Segments described by IMU orientations |
|---|---|
| 1 | Pelvis |
| 7 | Pelvis, thighs, shanks, feet |
| 12 | Pelvis, thighs, shanks, feet, torso, upper arms, lower arms |

Note:
   IMU, inertial measurement unit; MS, musculoskeletal.

W550 Power: Windows 10 Education 64-bit, Intel Core i7-6700 3.40 GHz 8-CPU processor, 32768 MB RAM). The laptop computer was used to examine if the software can perform sufficiently using portable devices that allow the measurements to be performed outside a laboratory environment. The desktop computer was used to examine how large difference the increased computing power of a typical desktop compared to a typical laptop makes on the performance of the software.

Execution times with computer-generated IMU orientations were measured with random unit quaternions. The quaternions represented random and sometimes unrealistic poses, which varied greatly between time frames and were time-consuming to calculate. Therefore, the measured execution times represent performance that is worse than in normal human motion, *i.e.*, they were a benchmark of unrealistically poor performance. The execution times were measured with data from different numbers of IMUs (one and seven for the lower body model and one, seven, and 12 for the full-body model). Each execution time measurement lasted until 10,000 IK operations were calculated. We report the mean, standard deviation and 95% confidence interval of the 10,000 execution times.

Finally, IMU orientations from real human walking were used to calculate throughputs and execution times. Performance was measured with pre-recorded quaternion orientations from 10 one-minute walking trials. Although the throughputs and execution times were calculated after the walking trials had been recorded, the performance tests were designed to simulate real-time measurement by feeding the quaternion data into the test environment one data frame at a time. The use of pre-recorded quaternion orientations enabled measuring throughputs above the sampling frequency of the IMUs; otherwise, the throughput would be limited to the sampling rate of the IMUs because IK operations could only be solved at the rate the IMU orientations are received. The tests were conducted with different numbers of IMUs (one and seven for the lower body model and one, seven, and 12 for the full-body model). Furthermore, different numbers of IK threads (one, two, four, six, eight) were used during throughput tests. The performance tests on real walking data allowed us to evaluate performance during a common human motion measurement.

## Error comparison of joint angles

Our software library includes an option to visualize motion like in OpenSim GUI by invoking methods from the Simbody API (namely, from the SimTK::Visualizer class), which OpenSim API is built upon. The use of the Simbody visualizer in a real-time IK

thread slows the thread noticeably, and if the throughput drops below the sampling frequency of the IMUs, the IK threads skip some orientation frames from the IMUs in a real-time measurement. This frame drop may negatively affect the accuracy of gait parameters derived from the IK solution. Although not a core feature of our software library, the Simbody visualization during RTIK may be of interest of some. Thus, we evaluated the effect of the frame drops on a kinematic variable that is often of interest and potentially affected by the frame drops, namely the ranges of motion (ROMs) of the joint angles. To this end we compared ROM between visualized RTIK (calculated in this case at approximately 45 Hz) and offline IK at the 60 Hz sample rate of the IMUs. Offline IK was calculated from IMU orientations that were stored after each of the 10 walking trials; RTIK with Simbody visualization was calculated on the desktop computer during each walking trial. Orientations from all 12 IMUs were included, and the full-body model was used in the analysis. The measured motion exerted 26 of the model's DOFs. For determining ROMs, the IK data was divided into periods based on the cyclical nature of the flexion-extension angle of the right knee. For each of the resulting 715 gait cycles (total from 10 walking trials), the difference between the highest and the lowest value of each joint angle was taken as its ROM. The results were reported as the mean absolute error (MAE) and 95% confidence interval (95% CI) between offline IK and RTIK ROM. Note that we calculated errors to evaluate the effect of visualization on ranges of motion and to demonstrate how the software library works, not to validate IMU-based IK.
For IMU-based IK validation, see other studies such as (*Al Borno et al., 2022*; *Tagliapietra et al., 2018*).

## RESULTS

### Performance tests

Performance test results showed that increasing model complexity and the number of IMUs for orientation tracking increased execution times (Fig. 2) and decreased IK throughput (Fig. 3). Increasing the number of IK calculating threads increased throughput (Fig. 3). The desktop computer always had lower execution times and higher throughput than the laptop computer in the same performance tests (Figs. 2 and 3).

Using computer-generated random unit quaternions, the means and standard deviations of execution times (operation delays) increased with increasing number of IMUs similarly on both the desktop and the laptop (Table 3, Fig. 4). With one IMU, the full-body model was 60–65% slower than the lower body model and had more variation in execution times. The mean execution times were approximately 25% longer on the laptop than on the desktop.

Execution times calculated from human walking (Fig. 2) are shorter than the corresponding execution times from computer-generated data (Table 3, Fig. 4) and remain below 55 ms even with 12 IMUs and the full-body model. Performance with real walking data follows the same patterns as with computer-generated data. Execution times increase (Fig. 2) when the number of IMUs or model complexity (number of DOFs) is increased.

Execution times below 30 ms using the laptop and below 25 ms using the desktop were reached with the lower-body model and seven IMUs (Fig. 2). Furthermore, even with the

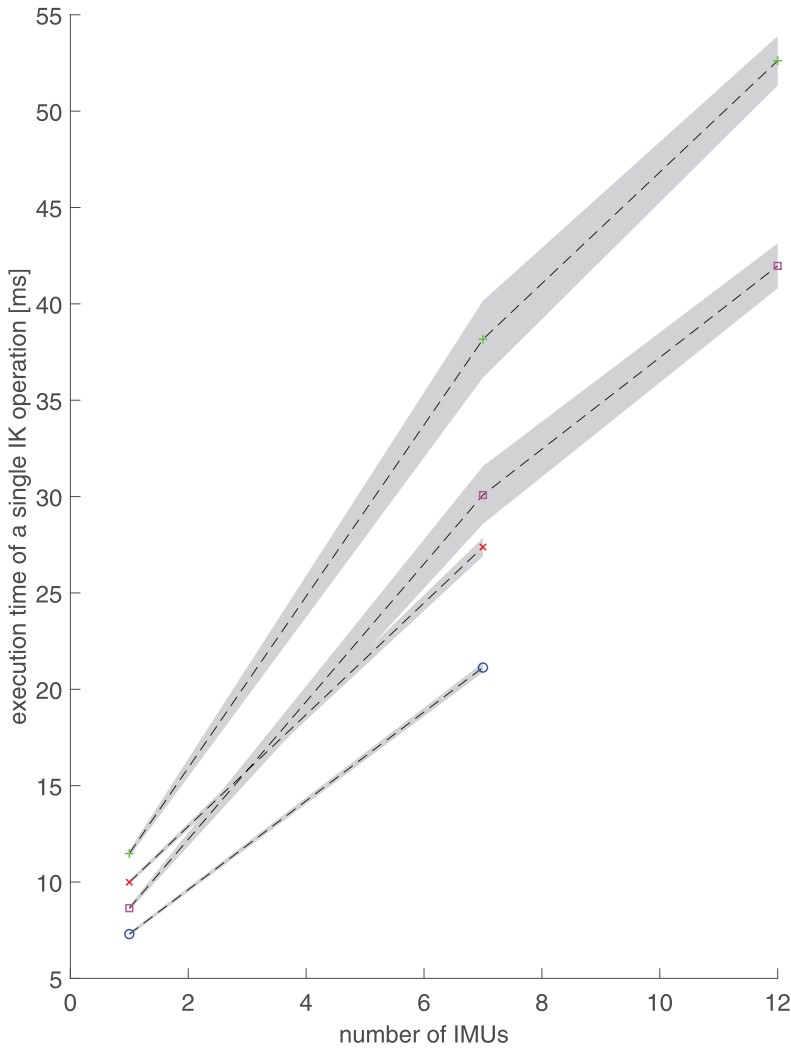

**Figure 2 Execution times and standard deviations (shaded area) of a single inverse kinematics operation with respect to the number of inertial measurement units (IMUs).** The execution times are presented as mean over 10 trials for two different musculoskeletal models (lower body and torso model and full-body model) using one, seven, or 12 IMUs. IMU quaternions orientations were retrieved from previously recorded walking trials. Red diagonal cross: lower-body model and laptop computer. Green cross: full-body model and laptop computer. Blue circle: lower-body model and desktop computer. Purple square: full-body model and desktop computer. 

laptop, full body kinematics can be calculated with execution times below 60 ms, and provided at least four IK threads are used, with throughputs above 50 Hz (Fig. 3).

 Throughput tests show that on the laptop, human walking can be solved at more than 60 IK operations per second when using eight IK threads, the full-body model and 12 IMUs (Fig. 3). The minimal IMU setup to record motion of all lower-body DOFs, seven IMUs with the lower-body model, reached a throughput of 130 with eight IK threads. On the desktop, the corresponding throughputs were 90 and 180, respectively.

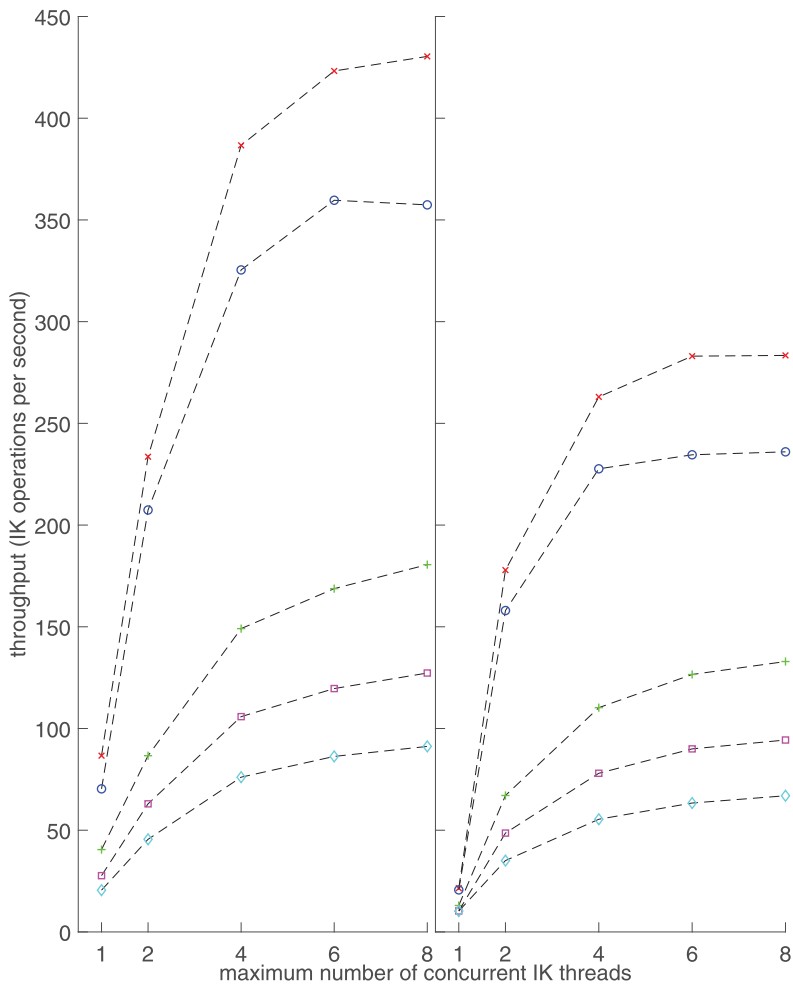

**Figure 3 Inverse kinematics (IK) throughput with respect to the number of IK threads used, measured on a desktop computer (left) and a laptop computer (right).** The throughputs are presented as mean over 10 trials. The measurements were repeated with two different musculoskeletal models, and using one, seven or 12 inertial measurement units (IMUs). IMU quaternion orientations were retrieved from previously recorded walking trials. Red diagonal cross: lower-body model and one IMU. Green cross: lower-body model and seven IMUs. Blue circle: full-body model and one IMU. Purple square: full-body model and seven IMUs. Cyan diamond: full-body model and 12 IMUs.

**Table 3 Mean, standard deviation (STD) and 95% confidence interval (CI) of execution times of a single inverse kinematics (IK) operation.**

|  | Lower body, 1 IMU | | Lower body, 7 IMUs | | Full body, 1 IMU | | Full body, 7 IMUs | | Full body, 12 IMUs | |
|---|---|---|---|---|---|---|---|---|---|---|
|  | Desktop | Laptop | Desktop | Laptop | Desktop | Laptop | Desktop | Laptop | Desktop | Laptop |
| Mean time (ms) | 6.97 | 8.52 | 43.55 | 56.96 | 11.52 | 13.77 | 48.12 | 64.01 | 80.42 | 95.24 |
| STD (ms) | 2.31 | 3.62 | 22.39 | 27.19 | 6.08 | 7.05 | 34.03 | 46.36 | 29.49 | 42.00 |
| 95% CI (ms) | 0.05 | 0.07 | 0.44 | 0.53 | 0.12 | 0.14 | 0.67 | 0.91 | 0.58 | 0.82 |

**Notes:**
The values are calculated over 10,000 IK operations for two different musculoskeletal models, two different computers and one, seven, or 12 inertial measurement units (IMUs). Randomly selected unit quaternions were used as IMU orientations.
CI, confidence interval; IK, inverse kinematics; IMU, inertial measurement unit; STD, standard deviation.

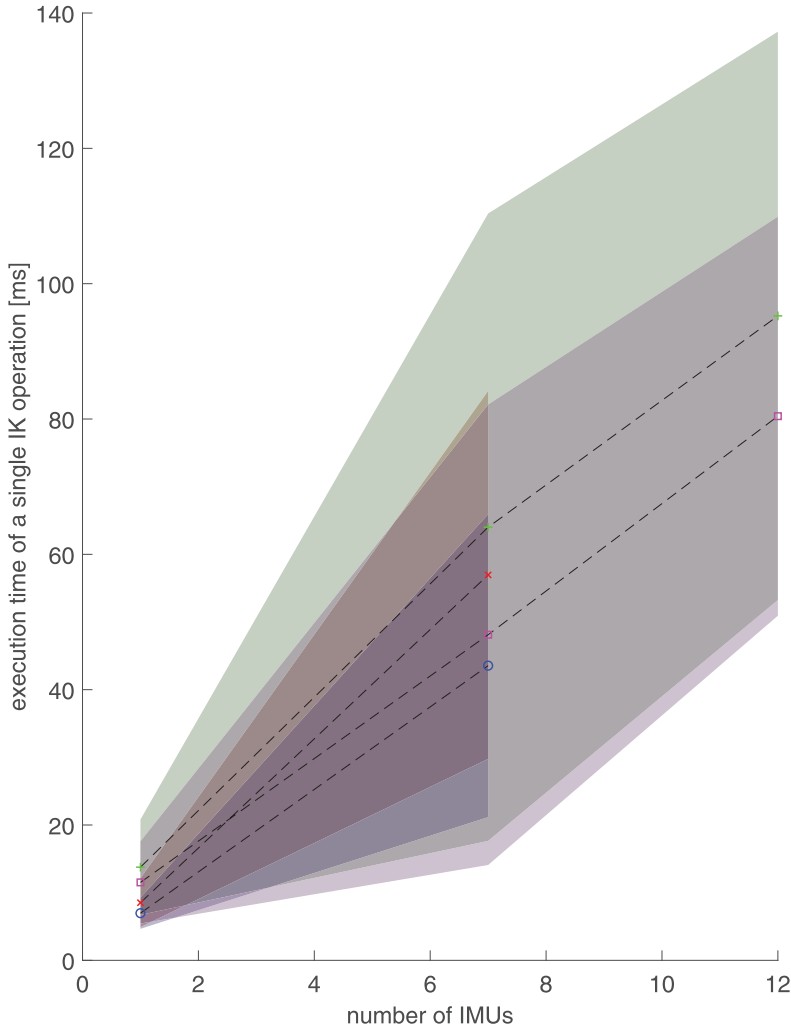

**Figure 4 Execution times and standard deviations (shaded area) of a single inverse kinematics operation with respect to the number of inertial measurement units (IMUs), *i.e.*, number of segments with random orientation data.** The values are calculated over 10,000 IK operations for two different musculoskeletal models, two different computers and one, seven or 12 inertial measurement units (IMUs). Randomly selected unit quaternions were used as IMU orientations. Red diagonal cross: lower-body model and laptop computer. Green cross: full-body model and laptop computer. Blue circle: lower-body model and desktop computer. Purple square: full-body model and desktop computer.

## Error comparison of joint angles

Enabling Simbody visualization during the measurement session reduced IK throughput but, compared to IK at full 60 Hz sampling frequency of the IMUs, caused only minimal differences in calculated ranges of motion (Fig. 5). The mean ROM error for all DOFs was 0.0675 degrees. The greatest MAE in ROM was observed in ankle joints (up to 360% of the mean for all joints), followed by the left hip joint (Fig. 5). Pelvis and upper extremities had the smallest ROM error. All MAEs remained below 0.3 degrees.
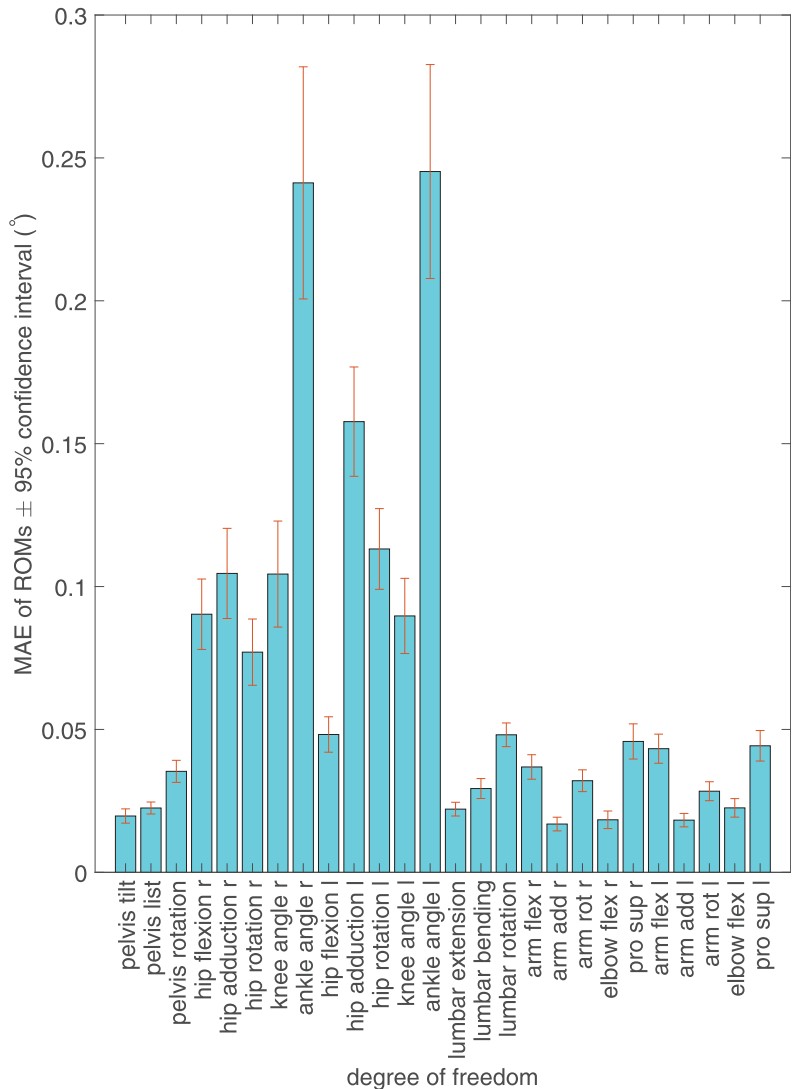

**Figure 5 Mean absolute error (MAE) between real-time inverse kinematics and offline inverse kinematics ranges of motion (ROMs) of the exerted degrees of freedom of the musculoskeletal model and the 95% confidence interval of the error.** Solid bars show MAEs of ROMs for each exerted degree of freedom. Confidence intervals are shown as error bars centered on the top of the MAE bars.

## DISCUSSION

We present an open-source software library for the real-time inverse kinematical analysis of IMU data with user-defined musculoskeletal models using OpenSim 4.1. Full-body IK can be calculated for random orientations in less than 100 ms; using real walking data, it can be done in less than 60 ms. On a desktop computer, the software library can solve RTIK at 180 samples per second while tracking the pelvis and lower extremities and at 90 samples per second while tracking the full-body kinematics. On a laptop computer, the corresponding throughputs were 130 and 60 samples per second, respectively. Using 12 IMUs to track walking and visualizing the results on a full-body running model, RTIK was solved at 45 samples per second on a desktop computer. The drop from the IMU output

sampling rate of 60 Hz resulted in a minimal difference in calculated joint ROMs (<0.3 degrees). The software library allows the use of RTIK virtually without limitations due to location or environment, which opens possibilities for a variety of applications including rehabilitation, ergonomics, and human-machine interfaces for controlling collaborative robots. Observing the movement of interest in a natural environment is important because a laboratory setting may affect how a person moves (*Friesen et al., 2020*).

## Performance tests

We investigated the execution times and throughputs of the IMU-based IK to determine if the output can be considered real-time. *Pizzolato et al. (2017)* used an execution time of 75 ms as the threshold for a real-time system. It was based on a study by *Kannape & Blanke (2013)* in which the subjects were able to identify the displayed motion as self-generated in real-time in over 80% of the cases if the delay in motion display was less than 75 ms. Even with a delay of 210 ms, subjects identified the visualized motion as self-generated in real-time in 50% of the cases. *Borbély & Szolgay (2017)* noted that the IK algorithm of OpenSim 3.3 had an execution time of about 145 ms, thus calculating IK at about 7 Hz and "falling behind the generally accepted practice in human movement recording of at least 50 Hz". Therefore, a real-time application should achieve IK throughput of 50 operations per second with an execution time below 75 ms for any single operation. With our software library, we aimed to achieve this target by using multithreading and the IK algorithm of OpenSim 4.1.

Another interesting finding by *Kannape & Blanke (2013)* was that subjects modulated their stride based on the delay between the motion and its visualization. Therefore, it is important to minimize the delay when preparing a real-time measurement setup to prevent subjects from altering their gait characteristics based on delayed visual feedback.

Live visualization is unnecessary in applications where IK is an intermediate output that is used to estimate contact forces, instruct a robot arm in rehabilitation applications or calculate gait parameters, to name a few examples. Thus, the performance tests were designed so that they evaluate only the performance of IK, which is the core feature of the software library. Although we adopt the visualization-based 75 ms criterion for real-time motion from Kannape and Blanke, our performance tests were conducted without visualization. Our software library relies on Simbody visualization and lacks an elegant visualization solution of its own, which is a limitation that should be acknowledged.

The performance tests show that IK throughput is more sensitive to the number of IMUs than the DOFs of the model, although model complexity also increases computational load because joint angles with no experimental data to solve them are still considered in the IK algorithm. Execution times clearly increased with the number of IMUs, although model complexity also affected them noticeably. Therefore, minimally enough IMUs and DOFs should be chosen to enable high throughput and short execution times in real-time.

The performance of the software library would benefit from improvements particularly in solving IK and visualizing the results. Because a new quaternion time series table is created for a single IMU data point every time an IK frame needs to be solved, the software

library is computationally much heavier than it would be if a single time series table could act as a permanent buffer that is updated with new IMU data points. This implementation was not possible using the OpenSim API during the development stage of our software library but appears to be supported in the latest version of the API. Furthermore, the data points in real-time IK may be solved in inconsistent order depending on how quickly each IK thread finishes. Expired IK frames may be omitted if the real-time IK is used as input in further real-time analyses but the missing data points may require interpolation or other consideration, *e.g.*, if filtering the IK is required. Thus, future development of the software library could implement existing interpolation functionalities from the OpenSim API. Additionally, if the visualizer would be implemented in its own thread, it could be used without blocking the IK solving threads and it would enable greater IK throughput while visualization is enabled. This improvement would also enable the solving of joint angles at a high frame rate and visualizing them at a reduced frame rate for performance reasons.

Finally, it should be noted that the software library was only tested on the Windows operating system, and the performance tests were conducted on its operation as a whole. Thus, no proper profiling analysis was done to discover which parts of the software library have potential for performance optimization.

## Execution times of the IK operation

Real motion, such as walking, contains a combination of different orientations, most of which are within a typical model's joint angle boundaries. Randomly generated unit quaternions used in execution time tests often result in unrealistic poses. As a result, the IK based on randomized unit quaternions is heavier to calculate than the average orientations during walking, or any typical human motion. Therefore, the execution times from computer-generated data can be interpreted as the worst performance when analyzing human motion without live visualization. Consequently, if the execution times with computer-generated data are sufficiently small for real-time analysis, then any realistic motion should be processed with smaller or equal execution times. On the other hand, we can assume that most human motions can be analyzed with execution times like those from real walking data because joints angles are likely to change at a similar rate and exert constraints similarly.

For both models using computer-generated random unit quaternions, the standard deviations of the execution times are on the same scale as the mean execution times, implying that there is great variation in the execution time. The randomized nature of the used quaternion orientations is a likely contributor to the high standard deviation, because randomized orientations occasionally lead to strange segment orientation combinations that do not reflect valid human motion and take the IK algorithm a long time to solve. This is further supported by the much smaller standard deviation in real walking data (Fig. 2).

Before conclusions are drawn from the computer-generated random unit quaternion-based execution times, it should be acknowledged that the results varied somewhat on repeated runs with the same parameters. This implies that running the tests more than 10,000 iterations could improve the precision of the results. However, because the purpose of the computer-generated random unit quaternions was to estimate

worst-case performance for the hardware of the tested computers and the effects of differing the number of DOFs of the model and segments with IMU orientations, which they show well, we chose not to repeat the tests with more iterations.

For both computer-generated (Table 3) and real walking data (Fig. 2), with one and seven IMUs, the execution times are shorter and vary less for the lower body model than for the full-body model. Both the mean execution times and the standard deviations are smaller on the desktop than on the laptop. However, the execution times vary less with 12 than with seven IMUs on the full-body model.

Because execution times calculated on real walking data remained below 55 ms (Fig. 2), the software library is capable of real-time inverse kinematics analysis of the full body even on a laptop. Using computer-generated random unit quaternions, the 95% confidence intervals of execution times are roughly 1% of the mean execution time in all cases, meaning that the execution times stay consistently below 75 ms except when 12 IMUs are used. In that case, the execution times stay consistently below 100 ms, which is still less than half of the 210 ms delay that marks 50% confidence in perceiving motion as real-time (*Kannape & Blanke, 2013*). Therefore, while RTIK is clearly possible with normal walking, some complicated motions may result in longer execution times but could nonetheless be analyzed practically in real-time. Because the execution times represent the minimum delay from the orientation data retrieval to the moment we can visualize or further analyze the IK output, the number of IMUs in a real-time measurement should be chosen considering the delays that are acceptable for the application.

## Throughputs

Figure 3 shows that increasing the number of concurrent processor threads increased the throughput until about eight threads, which was the maximum CPU core number for both computers. Increasing the number of IK threads further had no meaningful effect on the throughput, which was also observed in an earlier study on RTIK (*Pizzolato et al., 2017*). The throughput plateau resulted from CPU utilization of the computer reaching 100% and is thus hardware dependent.

The increase in throughput by multithreading is especially large with a small number of threads and one IMU. For example, throughput increases from less than 25 to more than 150 when the number of IK threads increases from one to two on the laptop. Doubled computational capacity alone cannot explain the increase in throughput. The effect is also present on the desktop. Furthermore, the relationship between the throughput and the number of IK threads is clearly nonlinear whereas an earlier RTIK study found it almost linear (*Pizzolato et al., 2017*). No explanation for this phenomenon was found, but it should be addressed in the future development of the software library.

For one IMU and four or more concurrent threads, the lower body model with 23 DOFs had approximately 20% higher throughput than the full-body model with 29 DOFs. For seven IMUs, the lower body model throughput was 40% higher than that of the full-body model. Therefore, model selection has a noticeable effect on the performance of RTIK and the model with the smallest sufficient number of DOFs should be chosen to reach maximal RTIK performance.

The software library is clearly capable of calculating IK at a higher rate than the lower limit of 50 Hz named by *Borbély & Szolgay (2017)*, but requires multithreading to reach it with complex musculoskeletal models. For portable real-time gait measurements, a laptop should be able to achieve sufficiently high IK throughput when seven IMUs are used. We reached a throughput of 130 samples per second in such a scenario, although others' results will vary depending on the hardware of the laptop. Nonetheless, a throughput of 130 samples per second should be sufficient for most gait analysis applications.

## Error comparison

Because loss of frequency may lead to reduced accuracy in measuring sharp peaks in joint angles, joints where motion direction changes fast are likely to have high ROM error (Fig. 5). During walking, ankle flexion (ankle_angle_r and ankle_angle_l) undergoes fast changes, which explains why its ROM error stands out. However, because all ROM errors remain consistently below 0.3 degrees, the effect of the drop in visualized IK from 60 to 45 Hz on ROM is very small.

The ROM error of left hip adduction stands out because it is visibly higher than that of the right hip. The error is caused by an artifact in IMU signal that caused the left leg to be violently jerked to the right after the left toe-off phase. The artifact is probably caused by the distortion of magnetic fields near the ferromagnetic laboratory hardware, which the left leg was closer to.

## Comparison to other solutions for IMU-based RTIK using OpenSim

In the introduction, we briefly presented some existing solutions for real-time IMU-based IK. Two of them, OpenSenseRT (*Slade et al., 2022*) and the solution by *Stanev et al. (2021)*, utilize the OpenSim API like our software library. OpenSenseRT had comparable execution times and throughputs but to optimize performance, they loosened IK solver tolerance and simplified their musculoskeletal model by removing muscles and locking unused joints. Therefore, the performance results are not easily comparable to ours, where many joints remained unlocked despite not having experimental IMU data to solve them uniquely and muscles of the model were left as they are (although muscles are not part of IK calculations, we made preliminary and unreported observations that OpenSim's standard IMU-based IK throughput is slightly lower on a model with muscles compared with a model without muscles). OpenSenseRT is a good solution for IMU-based IK where the Raspberry Pi computing unit is carried with the subject, while our solution relies on a computer separate from the subject, limiting its applicability slightly but allowing higher performance due to numerous options of laptop hardware and requiring less work to use different musculoskeletal models. The solution by *Stanev et al. (2021)* allows the use of different musculoskeletal models easily. They support both marker-based and IMU-based IK and use lower-level API functions to calculate IK quickly. Furthermore, their solution goes beyond IK, enabling even real-time inverse dynamics and joint reaction force analysis. Their software architecture relies on two threads: one to collect orientation from IMUs or marker positions and then perform IK, and another to perform preprocessing and further musculoskeletal analysis in real-time. Our solution has one thread for collecting

orientation from IMUs and a user-defined number of threads for IK. Therefore, although the solution by *Stanev et al. (2021)* is superior to ours in terms of number of features, our solution may allow higher throughputs due to the variable number of IK threads that is only limited by computer hardware.

## CONCLUSIONS

An open-source software library that builds upon the widely used OpenSim software was developed and published for IMU-based RTIK. It allows the joint angles of any OpenSim-compatible musculoskeletal model to be analyzed in real-time. While another real-time solution was concurrently and independently developed by *Stanev et al. (2021)*, its IK calculation does not utilize multithreading, which may limit its throughput, although its IK calculation relies on lower-level API classes that are faster than those used by the software library developed in this study when a single thread is used. The authors encourage others to contribute to the open-source project. The development of the software library will closely follow the development of OpenSim to utilize its built-in functionality for processing live data. The software library could be utilized in real-time estimation of joint moments, muscle forces, and joint contact forces based only on IMU data. Ground reaction forces and moments and kinematics are required for solving the equations of motion for the musculoskeletal model using inverse dynamics. It has been shown that ground reaction forces and moments can be predicted from IMU-derived kinematics (*Karatsidis et al., 2017*; *Stanev et al., 2021*). Moreover, estimation of muscle forces using optimization techniques uses kinematics and inverse dynamics estimates of joint moments as inputs and estimates of joint contact forces can be derived based on kinematics, inverse dynamics, and muscle forces. Hence, IMUs could be potentially used for the real-time estimation of musculoskeletal dynamics outside the laboratory and implemented in the software library in the future. Another interesting future application is the use of RTIK output together with EMG. Thus, combining IK output with EMG in real-time may provide interesting possibilities for estimating muscle forces and musculoskeletal loading using EMG driven musculoskeletal simulations (*Sartori et al., 2011*), for biofeedback to optimize rehabilitation or ergonomics or for biosignal-based operating systems.

### Funding

This work was supported by the European Union (European Regional Development Fund) and the University of Eastern Finland under the projects Human measurement and analysis research and innovation laboratories (HUMEA, project IDs A73200 and A73241) and Digital Technology RDI Environment (Digi Center, project IDs A74338 and A74340). LS received support from the Academy of Finland (#332915). The funders had no role in study design, data collection and analysis, decision to publish, or preparation of the manuscript.

## Grant Disclosures

The following grant information was disclosed by the authors:
European Union.
University of Eastern Finland (HUMEA): A73200 and A73241.
Digital Technology RDI Environment: A74338 and A74340.
Academy of Finland: #332915.

## Competing Interests

The authors declare that they have no competing interests.

## Author Contributions

- Jere Lavikainen conceived and designed the experiments, performed the experiments, analyzed the data, prepared figures and/or tables, authored or reviewed drafts of the article, and approved the final draft.
- Paavo Vartiainen conceived and designed the experiments, performed the experiments, authored or reviewed drafts of the article, and approved the final draft.
- Lauri Stenroth conceived and designed the experiments, authored or reviewed drafts of the article, and approved the final draft.
- Pasi A. Karjalainen conceived and designed the experiments, authored or reviewed drafts of the article, and approved the final draft.

## Data Availability

The source code of the software is available at GitHub: https://github.com/jerela/OpenSimLive.

The raw measurement data is available at Zenodo: Lavikainen, Jere, Vartiainen, Paavo, Stenroth, Lauri, & Karjalainen, Pasi. (2022). IMU data from walking trials on a treadmill (1.0.0) [Data set]. Zenodo. https://doi.org/10.5281/zenodo.5901448.

## Supplemental Information

Supplemental information for this article can be found online at http://dx.doi.org/10.7717/peerj.15097#supplemental-information.

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
