# Peer review of "Open-source software library for real-time inertial measurement unit data-based inverse kinematics using OpenSim"

_PeerJ, doi:10.7717/peerj.15097_

## Round 0.1 · original submission · Major Revisions

Both reviewers had significant concerns about the novelty of the work. Novelty per se is not required for PeerJ, but you do need to convince the reviewers that the work is useful.

Below are the relevant items from https://peerj.com/about/editorial-criteria/ :

* Decisions are not made based on any subjective determination of impact, degree of advance, novelty or being of interest to only a niche audience. We will also consider studies with null findings.

* Replication studies will be considered provided the rationale for the replication, and how it adds value to the literature, is clearly described. Please note that studies that are redundant or derivative of existing work will not be considered.

Please make sure to address all reviewer comments, point by point. Lack of novelty can be acceptable as long as your paper adds value to the literature.

·

Basic reporting

The language is well understandable. However, the language could be more concise (e.g., section about the fact that IK is not explained; section about tested models; section about error comparison) to explain more details precisely (e.g., calibration procedure) within the same length. For details, see my comments below.

The introduction is generally well written, but I am missing relevant literature. I agree with the authors that machine learning approaches have the drawback that domain knowledge is needed when the experimental setup should be adapted (line 81). However, in my opinion, the authors could motivate why an adaption of the sensor placement is needed. In literature, further machine learning approaches to estimate kinematics based on IMU data. E.g.:
=> Frank J. Wouda et al. “Estimation of full-body poses using only five inertial sensors: An eager or lazy learning approach?” Sensors (Switzerland) 16.12 (2016).
=> Frank J. Wouda et al. “Estimation of vertical ground reaction forces and sagittal knee kinematics during running using three inertial sensors”. Frontiers in Physiology 9.MAR (2018), pp. 1–14.

There also exist approaches using extended Kalman Filtering which are real time capable. E.g.:
=> Miezal, M., Taetz, B., & Bleser, G. (2017). Real-time inertial lower body kinematics and ground contact stimation at anatomical foot points for agile human locomotion. Proceedings - IEEE International Conference on Robotics and Automation, 3256–3263.

There was another parallel development for realtime IK which should be included especially since the preprint is available at ResearchGate since March 2021:
=> P. Slade, A. Habib, J. L. Hicks and S. L. Delp, "An Open-Source and Wearable System for Measuring 3D Human Motion in Real-Time," in IEEE Transactions on Biomedical Engineering, vol. 69, no. 2, pp. 678-688, Feb. 2022, doi: 10.1109/TBME.2021.3103201.

Hence, it does not get clear why the presented software package is required. Furthermore, I am missing the motivation why it might not be sufficient to stream the joint angles computed within Xsens to external software or why the proposed approach might be superior.

The overall structure matched standards. Nevertheless, clarity could be improved by rearranging the content of single subsections. In my opinion, methods on the evaluation are not structured perfectly. It might, for example, confuse the reader to first explain the execution time of a single IK operation (line 174-183) and the tested models (line 185-196) within the subsection “Execution times” and afterwards explain the throughput of the IK (line 200-218) within the subsection “Throughputs” which uses again the same models. I would recommend moving the explanation of the tested models including the IMU sets before the execution times, i.e., before the explanation of all the measures. Similarly, the subsection “Error comparison of joint angles” switches between measurement setup (e.g., treadmill, number of IMUs, protocol) and analysis (e.g., comparison of ROM, evaluated frame rate). Finally, the structure of the discussion should be improved.

The figures are partly blurred and should be provided in better quality.

Experimental design

The research is within the scope of PeerJ and in my opinion despite the other parallel developments of real-time IK still relevant. However, as stated above, the authors should elaborate more on the existing research and the relevance of their work.

In “Working principles of the software” (line 117-170), it is not explained whether a sensor-to-segment alignment is performed within the software. Therefore, it remains unclear till the end of the methods section (line 244) whether the authors rely on the IMU system calibration (Xsens or Delsys), or if no calibration is performed at all. I would recommend mentioning the calibration procedure already earlier since this is a crucial step. I am further wondering how the calibration procedure is included into the workflow: Is the calibration performed in real-time? Does it take the first seconds of streaming by default for calibration, or how is the calibration procedure started?

With respect to the analysis of the execution time and throughput (line 185-191 and line 204-207), I doubt that the solution of the IK problem (i.e., the optimization) is uniquely defined when using less IMUs than segments. Therefore, I am wondering whether a low number of IMUs are appropriate and allow a meaningful conclusion.

It is not clear to me why execution times were evaluated using random quaternions and the throughputs using identity quaternions. Additionally, I am wondering why real motion data was only used for the error comparison of joint angles, but not for the performance evaluation. The authors should elaborate more on these design choices. Were the random quaternions ensured to result in joint angles which are within the ROM of the model? Furthermore, the fact that identity quaternions were the calibration pose should be explained within the method section and not only in the discussion (line 354).

Moreover, the authors could further elaborate on the live visualization. It only gets clear that the software provides this feature when reading between the lines (line 217). For an application with real-time feedback, it would be important to evaluate the execution time and throughput with enabled visualization. Although it is stated that performance drops with enabled visualization (line 222), I would recommend presenting the execution times and throughput also for this condition in detail in the result section.

In my opinion, it should be mentioned in the main paper that no scaling of the model is necessary since sensor orientations and not raw inertial signals are tracked.

In the paper, it remains unclear how the ROM was determined. This should be explained within the methods

Validity of the findings

The motion data as well as the result from IK was provided for the error comparison of joint angles. I took a rough look at the data (trial 1 only) which rise some concerns. I understand that the authors don’t evaluate the performance of the IK based on IMU orientations in comparison to optical motion capturing. However, the analysis “error comparison of joint angles” might require more attention. The reconstructed motion does not look natural since legs are crossing (high hip abduction/rotation). This fact is shortly discussed in line 437-440, but only related to the left hip adduction while further angles might be influenced as well. Furthermore, the knee angle runs into saturation. A comparison of the ROM for saturated values is not very reliable. I also do not understand why the first 10 seconds vary considerably between live and offline processing.

In line 328-345, the authors compare their performance to the one of Pizzaloato et al. based on the definition of real-time of Kannape and Blanke. However, the definition of real-time is based on an experiment including visualization while the presented performance measures do not include visualization. This point is very shortly addressed in line 240/350 but should be made clearer and a performance evaluation including visualization would be desirable. Also in line 382-388, it should be stated clearly that visualization will also require a considerable amount of time. Hence, the overall performance appears too positive.

In line 375-378, it was stated that more than 10.000 iterations should be evaluated, but that this would take too long as 10.000 iterations take up to 20 minutes. In my opinion, it is reasonable to spend way more than 20 minutes on the performance test.

In line 415-419, the effect of the number of IMUs on the throughput is discussed. Do the authors think that a reason for the small effect might be the fact that the optimization problem is purely defined when using few IMUs and therefore harder to solve?

In line 421-426, the possible sampling frequency is discussed. However, it remains unclear if this conclusion was drawn from the execution times or throughputs. In my opinion, the throughput test does not reflect well a motion capture scenario since IMU data neutral standing was used which is close to the neutral pose of the model and therefore to the initial guess of the optimization (IK).

In line 446-450, the difference to Stanev et al. was discussed. Nevertheless, in my opinion, the common points and disadvantages should be discussed in more detail. Here, a comparison to Slade et al. (see above) should be added.

Additional comments

- Line 46 and 53: I would recommend putting references into the sentence instead of having it after the sentence
- Line 58: It probably should be “an IMU”
- Line 121: I would recommend to directly provide the link to github in this line.
- Line 200: This sentence is in my opinion not formulated precisely enough. It is not straight forward to understand that the throughput was calculated as average over one minute.
- Line 243 and 247: What is the difference between “Online Resource 1” and “Supplemental Article S1”? This should be presented more clearly.
- Line 268: Consistently use the abbreviation STD.
- Line 305: There is a typo in “90\%”

Reviewer 2 ·

Basic reporting

This manuscript presents the development of an open-source software library to estimate the human pose using inertial measurement units (IMU). The presented library leverages the capabilities of the popular biomechanics software OpenSim, therefore using an inverse kinematics algorithms to minimize the error between measured orientations (from IMUs) and model orientations. The performance of the developed library was assessed on two different musculoskeletal models, demonstrating the capability of the software to enable real-time execution.

Overall, the structure and the language of the manuscript could be improved. I have found multiple shorts and sometimes poorly connected paragraphs, and some unclear sentences throughout the paper. .Additionally, the proposed benchmarking appears to have some methodological flaws that may impact the reported results.

Experimental design

Developing software as part of research work is often very difficult and, despite the large amount of effort and time invested, often difficult to capitalize immediately. The authors have clearly dedicated much time and effort to the development of this software, as also demonstrated by the number of commits submitted to the online repository. However, while this is commendable work, I struggle to see how this research fills a well identified knowledge gap.

Here I would like to summarize the existing literature regarding real-time pose estimation in biomechanics. I will limit myself to the paper that were cited in the manuscript, as they largely overlap with my knowledge of the literature as well. In brief, van den Bogert at al., (2013) developed the first biomechanics real-time pose estimation system based on motion capture data. The novelties were the real-time calculation of multi-dof inverse kinematics, inverse dynamics, muscle forces via static optimization, and the small computational costs. Some limitations were the limited musculoskeletal model personalization, closed source, and no availability of IMU input (IMUs were not as good and popular as they are now). Pizzolato et al., (2017) enabled inverse kinematics and inverse dynamics in OpenSim (i.e.., possibility of personalization), optimized the filtering to minimize time delay, and introduces concurrent programming (e.g., producer/consumer, thread pooling) to remove idle times associated to input/output operations and increase throughput. While the software is open source, it is poorly maintained and not compatible with OpenSim 4.x, and therefore lacks IMU input. More recently, Stanev et al., (2021) published their real-time system based on OpenSim/Simbody. Their novelty was the introduction of IMU input to solve inverse kinematics, also enabling inverse dynamics and joint reaction force estimate in absence of ground reaction force.

While I am really appreciative of the authors effort in developing the proposed software, bur when contextualized together with the current literature, I really struggle to see how the proposed solution improves upon previous publications in this area. My suggestion would be paring this work with an appropriate application, where the developed features could be clearly highlighted.

Validity of the findings

I have several comments on the proposed method, which might have consequences on the reported findings.

- L 143-145, I am unclear what it means that “The orientation information from IMUs is combined in a time series table that contains only one sample, i.e., time point. The time series table is given to OpenSim’s IK solver object, which solves the IK for that time point.”. Does it mean that every time an IK frame needs to be solved, an new object of type OpenSim::TimeSeriesTableQuaternion is created? This is extremely inefficient use of resources and would impact on performance.

- L 153-158. This section seems to imply that every time a single frame of IK is created, a new thread is also launched. Again, this is extremely inefficient. Creation of new threads is costly and should be avoided when possible. Ideally, thread that perform a specific computational task should be reused, hence avoiding the creation/destruction of new thread, where data is instead passed to the existing thread following producer/consumer or publisher/subscriber design patterns. See the thread chapter of Tanenbaum, A. (2009). Modern operating systems. Pearson Education, Inc.,.

- L 166. The time reconstruction of solved model poses (from the thread pool) appears to be sorted only at termination of the program. This seems unusual, as the live stream may result scrambled up, which obviously would be incorrect.

- L 175. It is unclear how the time measurement was performed exactly. What exact functions where used?

- L180 – 183. Why not using real data instead of random quaternions? IK in OpenSim is implemented using a gradient descend optimiser, where the first guess is based on the previously calculated joint angles, saved in the opensim state variable. So, by creating random quaternions you are actually not leveraging the fact that different IMU data frames should have similar solutions, hence converge quickly.

- L 202 – 203. Similarly, the use of identity quaternions (i.e., zero rotation) will not solve any IK at al, because the IK optimizer is already at the objective function local minima. Thus, the throughput test, is artificially boosted, not representing the true throughput that might be expected in a real experiment.

- I do not understand the rationale behind testing the system on a laptop and a desktop computer.

- I do not understand the problem with the live visualization. Why does the data rate drop? To me, it seems an implementation issue. I haven’t checked the code, but from this sentence I imagine that the visualization is not in a thread that is independent from the IK solver, hence the frame drawing gets in the way of the solver, reducing the frame rate.

- L 244. Please describe exactly what IMU calibration procedures were followed, as OpenSim standards might change over time, and it is important to clarify what is being done in this particular situation.

- L 254. It is unclear the rational of comparing ROMs rather than a full time series analysis. I suggest performing appropriate analyses of time series (e.g., RMSE, R2, SPM, etc).

- L 404-406. This effect is likely due to the continuous creation of thread. Have you performed any profiling analysis on your code?

Additional comments

- A brief description of the inverse kinematics algorithm employed in OpenSim (including equations) should be reported for clarity.
- Why is Stanev et al (2021) discussed in the conclusion section? It should be extensively discussed in the introduction as well as used as contextualization of the presented results in the discussion.
- Finally, the structure and the language of the manuscript should be greatly improved (e.g., disconnected and short paragraphs, unclear portions of the methods, etc).

---

## Round 0.2 · Minor Revisions

Both reviewers recommended minor revisions, and the suggestions seem quite straightforward.

Please make sure that comment 2 of reviewer 1 is properly addressed, with changes in the manuscript.

·

Basic reporting

The authors revised the manuscript carefully and rewrote a couple of paragraphs. Most of my comments were well-addressed. The introduction now reflects the state-of-the-art better, and the structure of the method section was improved. However, two comments with respect to the introduction remained:
1. The authors have added additional literature to provide a better overview of the state-of-the-art. This will be very useful, especially for readers that are not aware of the state-of-the-art. However, in my opinion, the flow of the introduction could still be considerably improved to enhance readability.
a. For example, the focus between inverse kinematics based on marker data and based on IMU data switches back and forth.
b. I recommend moving the description of OpenSim (line 57-62) to the sentence where the authors are motivating the usage of OpenSim (line 87-89).
c. When starting to read the paragraph about the work of Stanev et al. (2021) and Slade et al. (2022) (line 98-107), it is not obvious why these two papers are presented separately from the other RTIK approaches. I would recommend highlighting the fact that they use the OpenSim API already in the first sentence of this paragraph to better combine paragraphs.

2. I thank the authors for the detailed explanation of the difference between XSens and OpenSim which they provided in the rebuttal letter. My original comment might not have been clear enough on this topic. I apologize for the inconvenience. I will try to rephrase my point on this topic: In my opinion, the contribution of the paper could be further supported by highlighting the differences between other (commercial) approaches (e.g., as applied in the XSens software) and the use of OpenSim. Sometimes it might be more convenient to use the joint angles computed by the software of the sensor provider (e.g., XSens, Vicon, etc.). However, as the authors mentioned, the interface to OpenSim enables the user to modify the musculoskeletal model. In addition, the usage of OpenSim enables further analysis of, for example, kinetics. Although the customizability of the models is mentioned as an advantage in the revised version of the paper, these points could be explained in more detail.
I agree with the authors that the computation of joint angles from IMU orientations without the constraints of a biomechanical model would be less accurate. However, XSens is also taking a biomechanical model into account when computing joint angles from IMU orientations (see Roetenberg, D., Luinge, H. J., & Slycke, P. (2013). Xsens MVN : Full 6DOF Human Motion Tracking Using Miniature Inertial Sensors. Journal of Biomechanics, 13(1), 155–158.). XSens was just mentioned here as an example, as the experimental data in this paper was captured with XSens sensors.

Experimental design

The authors for adding explanations with respect to the IMU calibration to the manuscript and supplementary article. This is, in my opinion, now well explained. I also highly welcome and appreciate that the performance tests are now also conducted with measured gait data. Nevertheless, I still have two comments which remained:

3. This relates to the comment I made on executing a performance test with a low number of IMUs. I agree with the authors that there are a lot of use cases where the number of sensors is smaller than the number of segments in the model. As it was still unclear to me how OpenSim will handle IK problems with sparse tracking information, I made a quick test by tracking a sparse marker set with Hamer’s model. OpenSim resulted, for example, in an elbow flexion varying around 75° when no marker at the shoulders and arms were used. This would correspond to the mean of the allowed range of motion and not to the default value (as the authors write in the rebuttal letter). However, I do not think that OpenSim simply uses the mean value since the joint angles, which were not described by markers, varied over time. Hence, OpenSim or the underlying SimTK code is performing some computations. Based on this, in my opinion, the sentence which was added in lines 226-227 does not provide sufficient information or is not precisely phrased. From the sentence, as it is stated now, it could be interpreted that it is an active design choice of the authors not to optimize those joint angles. Furthermore, it will still be unclear to the reader what happens to the other joint angles. In summary, I recommend rephrasing the sentence to give precise information.
This point also gets interesting when comparing the execution time and throughput of the two models. The computation time would not increase so heavily with the complexity of the model if the IK algorithm would “ignore” all degrees of freedom that are not determined by the IMU data. Based on the current discussion, it is unclear to the reader why the computation time increases considerably with the model complexity. Hence, I recommend discussing this topic.

4. This relates to the comment I made on the explanation of the ROM computation. I welcome the additional details on it. However, it might not be clear to the reader that the ROM was computed for all angles. I recommend rephrasing it a bit to be more precise by using an expression like “the difference between the highest and the lowest value of each joint angle”.

Validity of the findings

-

Additional comments

I have a couple of other (partly minor) comments that should be addressed before publication to further improve the quality of the manuscript:

5. Abstract/Discussion: Reported numbers for execution time have changed considerably since the last version of the manuscript (e.g., line 31: 55ms instead of 100 ms; line 32: 90 samples per second instead of 900 samples per second). I did not find an explanation for this change in the rebuttal letter and was wondering why those numbers changed. The original problem behind this is probably that it sometimes remains unclear from which test the reported numbers are. For example, it is unclear to which test the reported numbers in the abstract or lines 485-487 belong without checking the results or figures again. Hence, it is difficult to interpret the text. A more precise phrasing is needed here to make the text easier accessible.

6. Lines 34-35, 477, 495-496: Please give the unit for the throughput. Even though it is relatively clear that the unit is samples per second, a unit should always be reported when giving numbers.

7. Line 62: The formulation of this sentence is imprecise and should be corrected. During IK, the difference between IMU orientations and segment orientations, but not joint angles) is minimized.

8. Line 65 and 69: The abbreviation real-time inverse kinematics is defined twice. After defining the abbreviation once, the abbreviation should be used.

9. Line 69 and 75 and lines 516-517: When the author names are given in the sentence, the author names should not be given again in the brackets (line 69) and the author names should not be in brackets (line 75). The correct formatting would be “Bonnet et al. (2013)”. However, the citations might be adapted by the technical staff.

10. Lines 149-151: The other reviewer made a comment on these lines. I appreciate the answer the authors gave in the rebuttal letter but suggest discussing this in the discussion section to make readers aware of this possibility for improvement.

11. Lines 225, 247, and 260: It is a bit confusing that the Gait2392 model is called “lower body and torso model” in these lines even though, in line 222, it was explained that the Gait2392 model will be called “lower body model”. Hence, I recommend removing “and torso” in lines 225, 247, and 260.

12. Line 249: The authors stated in the rebuttal letter, as an answer to my comment to line 268, that they decided not to use the abbreviation of “standard deviation” outside of captions. Either using the abbreviation in the main text consistently or not using it at all is fine. However, when not using it in the main text, I recommend removing the definition of the abbreviation from the main text in line 249.

13. Line 289: The OpenSense paper of al Borno et al. was published in February 2022. Therefore, the reference should be updated.

14. Table 3: I recommend calling the models “lower body” and “full body” as you did in the text and, for example, in Figure 2 and 3. When naming the models consistently, I also do not see a need anymore to call the models “Hamner” and “Gait2392” in lines 303-304 and 450-451.

15. Figure 2 and Table 3: It would be nice to compare the tests with the random quaternions and the measured gait data more easily. The authors could also provide graphs for the test with the randomly selected quaternions.

16. Lines 334-337: In the text, performance values are reported without specifying the maximal number of concurrent IK threads which was used for these values. Hence, the reader cannot make use of the numbers provided in the text without double-checking them in Figure 3. The number of maximum threads should also be mentioned in the text.

17. Lines 441-443: I think that the first part of the sentence is missing a verb. For example, it should be “Before conclusions are drawn from …”.

18. Line 503: There should be a comma after “During walking”.

19. Lines 499-506: I understand that visualization is not the main goal of the proposed library, as it is part of SimTK. However, the library could, in the future, support the visualization of the computed joint angles at a reduced frame rate while computing the joint angles at the original frame rate for later analysis. With this feature, the accuracy of the computed joint angles would not suffer from the visualization. This idea could be added to the discussion.

20. Lines 517-519: This sentence implies that removing muscles reduces computational complexity. However, it is not clear why removing muscles from the model file reduces complexity since muscles are not used in IK anyways. Furthermore, locking unused joints is, in my opinion, a better solution than letting the IK algorithm select some value. Could the authors please elaborate in the discussion more on these options, which the users could also take when using the proposed software library?

21. Supplementary Article - IK algorithm: Based on the comment of the other reviewer, an explanation of IK was added in the supplementary article. I highly welcome this addition. However, the variables used in the equation were not explained correctly. The generalized coordinates are usually in OpenSim global positions and orientation of the pelvis (or another segment) and joint angles. N is here not the number of generalized coordinates but the number of IMU orientations/sensors that are tracked in the IK algorithm. Accordingly, the index i of w_i and x_i is referring to the i’th IMU orientation/sensor and not to the i'th generalized coordinate. Furthermore, I recommend adding a reference to the supplementary article in line 135 in the main manuscript to actually point the reader to the additional information.

Reviewer 2 ·

Basic reporting

I would like to thank the authors for answering my questions.

Experimental design

1) Regarding the time measurement, I would like to see mentioned std::chrono::high_resolution_clock in the manuscript, as different standards exist to measure computational time.

2) Instead of implementing your own visualizer, which I agree would be out of scope, why didn't you move the visualizer in a different thread? It would be a matter of half a day work and would greatly improve results for your biofeedback applications.

Validity of the findings

3) The choice of not sorting out the frames after the thread pool has numerous ramifications that you should consider. If your goal is only visualizing movement, I agree with you that skipping a frame here or there is not a big issue. However, since you are using OpenSim, you are likely to use the results of the IK to perform further operations, such as calculating moment arms or inverse dynamics, or even controlling a robot. However, you need to consider that if you skip a frame (because one of the threads in the pool was too slow), then you start having issues with filtering, as you will have inconsistent timestamps and traditional FIR or IIR filters will have problems with that. How would you consider solving that? I would like to see this specific limitation addressed in the discussion. Also, you could have used an additional thread to sort out the frame ordering, before moving the data to the visualizer.

---

## Round 0.3 · Minor Revisions

Thank you for addressing the reviewer comments. Attached are some minor edits and comments from the Section Editor. Please address those and resubmit your final revision as soon as possible.

---

## Round 0.4 · accepted · Accept

I have confirmed that the authors have addressed all comments. The manuscript is ready for publication.